# Sexual and Agency Norms: Effect on Young Women’s Self-Perception and Attitude Toward Sexual Consent

**DOI:** 10.3390/bs15030250

**Published:** 2025-02-22

**Authors:** María del Mar Sánchez-Fuentes, Antonio Rafael Hidalgo-Muñoz, Nieves Moyano, Carmen Gómez-Berrocal

**Affiliations:** 1Mind, Brain and Behavior Research Center, University of Granada, 18011 Granada, Spain; mmsanchez@ugr.es; 2Department of Basic Psychology, Psychobiology and Methodology of Behavioral Sciences, Faculty of Psychology, University of Salamanca, 37005 Salamanca, Spain; arhidalgom@usal.es; 3Department of Developmental and Educational Psychology, Faculty of Humanities and Education Sciences, University of Jaén, 23071 Jaén, Spain; mnmoyano@ujaen.es; 4Department of Social Psychology, Faculty of Psychology, University of Granada, 18011 Granada, Spain

**Keywords:** sexual double standard (SDS), sexual agency norm, sexual reputation threat, self-concept, sexual consent

## Abstract

Background: The sexual double standard (SDS) governs behaviors related to sexual activity and abstinence, promoting negative evaluations of sexually assertive women. Conversely, the sexual agency norm encourages young women to express their sexuality freely. This study explores how this complex normative context, combining SDS and sexual agency norms, impacts young women’s self-concept and attitudes toward sexual consent. Methods: A total of 154 Spanish university women (*M*_age_ = 19.69 years; *SD* = 2.23) participated in a 2 × 2 quasi-experimental design [Personal Reputation Threat: Sexual Activity vs. Sexual Abstinence × Agency Belief Affirmation: High vs. Low]. Participants completed a self-perception measure and the Spanish Adaptation of the Sexual Consent Scale-Revised. Results: Personal Reputation Threat influenced positive self-evaluation, with higher scores in the Sexual Activity (vs. Abstinence) condition. Agency Belief Affirmation also affected self-evaluation, with higher scores in the low-agency (vs. high-agency) condition. An interaction effect emerged on attitudes favoring sexual consent: participants showed greater support for sexual consent when devalued for sexual assertiveness rather than abstinence, but only under low-agency belief affirmation. These findings highlight identity conflicts and inconsistent sexual attitudes that young women may experience within the complex normative frameworks of contemporary Western societies.

## 1. Introduction

In Western democratic societies, women’s sexuality continues to be assessed against a standard that distinguishes between virtue, embodied by sexual abstinence and shyness (i.e., the “virtuous” woman), and promiscuity, marked by hypersexuality and activity (i.e., the “promiscuous” woman) ([4]) Gender norms influence the expected behaviors of men and women, particularly in relation to sexual conduct ([17]; [41]). When a young Western woman has to decide what the most appropriate sexual behavior is or what her attitude toward sexual consent should be, she faces a complex regulatory environment.

First, gender norms such as the sexual double standard (SDS) promote expectations that women (compared to men) should demonstrate greater passivity, submission, and less self-determination and sexual autonomy ([9]; [28]). Studies conducted with samples of Spanish speakers have shown that a gender double standard is associated with sexual behavior, both regarding the open expression of sexuality (i.e., sexual freedom) and modesty in sexuality (i.e., sexual shyness) ([2]; [3]; [21]; [46]; [51]). Sexual norms function as a mechanism of control, reinforcing gender stereotypes through socialization, media representation, and the penalization of deviations ([45]; [60]). For instance, the likelihood of experiencing humiliation or devaluation for one’s sexual behavior—commonly referred to as slut-shaming—tends to increase when a person’s physical appearance or sexual conduct (typically a woman’s) is perceived as signaling sexual availability or challenging gender stereotypes ([56]).

On the other hand, young Western women are encouraged to exercise agency—to become assertive initiators who challenge traditional gendered sexual norms. At this point, it is important to clarify the difference between the conception of agency as an element of women’s sexual subjectivity and the other conception in which agency is part of the normative context of society. Studies that have focused on sexual agency as a subjective characteristic have shown that if young women perceive themselves as sexual subjects with self-determination and autonomy, they may be less likely than other women to engage in unwanted sexual activities to please a male partner ([26]). When sexual agency is understood as a normative dimension ([4]; [5]), we then assume that, by the very fact of being a norm, it influences people’s ideas, values, and beliefs. It has been suggested that the norm of sexual agency promotes a permissive discourse around female heterosexuality ([12]) and encourages the liberation from external restrictions (e.g., submission to traditional gender roles), fostering the exercise of free will, and, consequently, personal responsibility for the outcomes of one’s own behavior ([19]). From this perspective, we pose the following question: if a woman feels sexually devalued by others, how will her self-image and self-concept be affected when she agrees with the idea that, in society, anyone can be an agent and have control over their own behavior?

It has been suggested that the agency norm and the norm regarding gendered sexual behaviors (sexual activity vs. abstinence and modesty) are independent dimensions that maintain an orthogonal relationship ([4]; [5]). We hypothesize that this normative environment, with its contradictory expectations (e.g., abstain–act, refuse–initiate, be sexually receptive–become an agent of one’s own sexuality ([18]; [32]) induces concern in women about being judged according to stereotypes associated with their social group ([53]). Female sexuality, unlike male sexuality, is regulated through sexual reputation ([12]). Consequently, the ambiguity of this normative context can undermine a woman’s reputation depending on whether she exhibits sexually uninhibited or restrained behavior. If a woman expresses sexually uninhibited attitudes and behaviors, the SDS norm may heighten the likelihood of her experiencing slut-shaming ([56]); furthermore, they may be viewed both as desirable sexual partners ([13]) and as promiscuous, easy, or selfish ([14]; [28]). Paradoxically, the agency norm may lead to a woman who presents herself as abstinent or sexually inhibited, being classified as sexually uptight or prudish ([58]).

The central aim of this research is to examine the impact of this normative environment, typical of contemporary Western societies, on young women’s self-image and their attitude toward sexual consent. To the best of our knowledge, this is one of the first studies to use a quasi-experimental methodology to examine the combined effect of sexual norms on young women’s identification with stereotypical traits and their attitudes toward sexual consent.

Following the previously established framework, a threat to one’s sexual reputation may emerge when others (i.e., society at large) negatively evaluates a woman’s behavior ([49]; [50]). In situations like this, the fear of confirming negative stereotypes may lead to the internalization of stereotypical expectations and roles, which can adversely affect women’s self-image ([23]; [52]). Research on stereotypes about women has shown that people develop schemas for different subtypes of women ([20]) and that the attributes used to characterize these subtypes align with the dimensions that determine the social status and value assigned to groups within the broader social structure ([20]; [27]). Overall, studies on stereotypes have demonstrated that the traits constituting stereotype content—whether applied to social groups or specific subtypes of women—are organized along dimensions that shape the perceived status and value of the group being categorized. These dimensions include competence and sociability ([15]; [14]), culture, and nature ([43]), as well as feelings and emotions ([57]). This study aims to examine whether, in a context of perceived threat to sexual reputation, women describe themselves using specific stereotypical dimensions. Within this framework, we propose the following hypothesis:

**Hypothesis 1.** 
*When a woman’s sexual activity is devalued or negatively judged by others, it can detrimentally affect her self-image ([23]; [52]). Specifically, it can affect or influence a woman’s tendency to identify with positive and negative traits associated with competence, sociability, nature, culture, feelings, and emotions.*


Furthermore, the perception that women have that their sexual behavior may be devalued by the context, can make heterosexual intimate encounters a threat to gender identity ([34]; [53]). In this regard, the normative context can influence a woman’s attitude toward sexual consent (SC), which represents the conscious willingness to engage in a specific sexual behavior with a specific person and within a specific context ([61]). Sexual consent is a complex concept involving internal states, explicit acts, and communicative behaviors. It requires a clear, voluntary, and coercive-free agreement, and it must be understood in the context of power dynamics and ongoing communication. Recent studies emphasize the importance of verbal and nonverbal cues to express consent ([16]; [24]; [25]; [39]). In Spain, sexual consent has shown increasing importance both in the legal sphere and in social perception, due in part to the 2022 reform of the Penal Code with the well-known Law on the Comprehensive Guarantee of Sexual Freedom, which includes the concept of sexual consent in defining sexual violence crimes. Therefore, it can be stated that it is not a taboo subject, and there is recent research ([37]; [38]; [36]). Within this framework, we propose the following hypothesis:

**Hypothesis 2.** 
*When a woman’s sexual activity is devalued or viewed negatively by others, it may influence her attitude toward sexual consent.*


It has been suggested that situations involving stereotype threat ([53]) induce dissonance in individuals, as they experience an imbalance among various elements: the stereotype of their social group (e.g., “women are not expected to be sexually assertive”), the area of capability in question (e.g., “I am a sexually assertive woman”), and self-concept (“I am a woman”) ([47]). According to this model, interventions to counter stereotype threat aim to foster beliefs that can alter the imbalance among these dissonant elements, for instance, by changing one’s beliefs about the stereotype ([31]). As previously noted, the norm of agency promotes freedom from external restrictions (i.e., traditional gender sexual roles) ([4]). Therefore, we hypothesize that the threat to sexual reputation arising from social evaluation may have a different effect depending on whether a woman believes her degree of agency is high (e.g., “sexual behavior depends on individual personality”) or, conversely, she believes her agency is limited (e.g., “as a woman, I am expected to meet certain expectations”). In this context, it is worth asking (RQ1) to what extent beliefs about agency (i.e., high agency vs. low agency) may moderate the threat to a woman’s sexual reputation. In developing our hypotheses, we first consider that the agency norm reinforces the belief in free will and personal responsibility ([4]). Second, we assume that a person’s self-concept may be affected depending on the type of cause to which they attribute the outcome of their behavior (i.e., an external cause vs. an internal cause related to a personal trait) ([59]). From this perspective, a woman’s perception of negative judgments about her sexual behavior can vary based on her level of agreement with the beliefs upheld by the agency norm. When faced with such evaluations, she may attribute the devaluation either to her own personal characteristics (indicative of high-agency beliefs) or to broader gender stereotypes (reflecting low-agency beliefs). Based on this approach, we propose the following hypotheses:

**Hypothesis 3.** 
*When the belief in ample freedom to exercise free will is reinforced—high-agency beliefs (vs. when one believes that sexual behavior is regulated by expectations or gender norms: low-agency beliefs)—self-image will be more negatively affected.*


**Hypothesis 4.** 
*When the belief in ample freedom to exercise free will is reinforced—high-agency beliefs (vs. when one believes that sexual behavior is regulated by expectations or gender norms: low-agency beliefs)—the attitude toward sexual consent will be more negatively affected.*


**Hypothesis 5.** 
*An interaction effect is expected to be found between threat to personal reputation and beliefs about agency ([47]).*


Finally, certain personal characteristics may influence an individual’s response when faced with a situation that threatens one’s sexual reputation due to gender norms and stereotypes. Firstly, the more a person identifies with the group that is negatively evaluated, or the more they expect to be perceived as a member of that group, the greater the stereotype threat they will experience in situations where a negative stereotype is applied ([53]). In this regard, we ask two questions: what role does the centrality of gender identity play in women’s responses? (RQ2) and what role does their gender self-esteem play? (RQ3). Secondly, individuals who hold a neoliberal ideology are more likely to support and conform to the agency norm ([6]). From this framework, we ask: What role does women’s adherence to neoliberal ideology play? (RQ4). Lastly, an individual’s perception of norms, such as those related to gendered sexual behaviors and the sexual double standard (SDS) ([22]), directly influences the willingness to endorse stereotypes ([7]; [48]). Therefore, it is important to analyze the role that perceived social support for gendered sexual norms and the SDS plays in women’s responses when they perceive their sexual reputation to be devalued (RQ5).

## 2. Materials and Methods

### 2.1. Design

The study followed a quasi-experimental, between-groups design (2 × 2) with two independent variables. The first manipulated variable was Threat to Personal Reputation (due to sexual activity vs. sexual abstinence), and the second was Affirmation in Agency Beliefs (high-agency beliefs vs. low-agency beliefs). The dependent variables included identification with 32 characteristics corresponding to eight dimensions (i.e., competence, sociability, sentiment, emotion, nature, culture, collectivism, individualism) and attitude toward sexual consent. Covariates considered in the manipulation included gender identity centrality, gender-based self-esteem, neoliberal beliefs and values, and perceived societal support for gendered sexual norms and the sexual double standard.

### 2.2. Participants

A total of 202 participants accessed the questionnaire link. Thirty participants were excluded because they did not affirmatively respond to the experimental manipulation check regarding Threat to Personal Reputation (see Section 2.3). An additional 18 participants were excluded for not responding to the manipulation check question on Affirmation in Agency Beliefs (see Section 2.3). The final sample consisted of 154 adult female Spanish undergraduates (*M*_age_ = 19.69 years; *SD* = 2.23). Among them, 43.5% reported being in a romantic relationship, and 79.1% identified as heterosexual. The distribution of participants across experimental conditions for multivariate analyses was as follows: Threat for Sexual Activity—low-agency beliefs (*N* = 34), Threat for Sexual Activity—high-agency beliefs (*N* = 44), Threat for Sexual Abstinence—low-agency beliefs (*N* = 40), and Threat for Sexual Abstinence—high-agency beliefs (*N* = 36). In cases where specific items were left unanswered, data were not extrapolated. Instead, missing responses were excluded from the analyses affected.

### 2.3. Procedure

The study was conducted according to the guidelines of the Declaration of Helsinki and approved by the Ethics Committee on Human Research of University of Granada (reference n. 3988/CEIH/2024). Data collection took place in a classroom setting. Each experimental session included up to 25 participants, who were welcomed by a female researcher trained in administering the study protocol. Participants were informed that the study aimed to understand how self-perceptions or self-images are shaped when individuals receive information about their sexual profiles or behaviors. To enhance the credibility of the experimental manipulation, participants were told that their responses would contribute to databases used by Artificial Intelligence (AI) programs to develop personal profiles.

Participants completed the assessment using the LimeSurvey platform, accessed via a QR code. Four distinct QR codes were created, one for each set of questions corresponding to the four experimental conditions. Using incidental non-probability sampling, each participant was assigned one of the QR codes. Upon accessing the application, participants signed an informed consent form before responding to the study measures. The sequence of instrument presentation and manipulations was structured as follows: first, participants completed scales measuring variables treated as covariates in the analyses.

Subsequently, the manipulation of Threat to Personal Reputation was introduced. Participants were provided with hypothetical sexual profiles based on their responses to the survey up to that point. Half of the participants were informed that their responses aligned with the profile of a sexually active and assertive woman (condition: Threat to Personal Reputation for Sexual Activity), while the other half were informed that their responses aligned with the profile of a sexually reserved and abstinent woman (condition: Threat to Personal Reputation for Sexual Abstinence). In both cases, participants were told that society generally holds a negative view of the corresponding sexual profile. To ensure that participants had read the provided information about their assigned sexual profile, a verification question was included.

Following this, the manipulation of the second independent variable was introduced (Affirmation in Agency Beliefs). Half of the participants were asked to write beliefs affirming the existence of sexual agency for women (high-agency beliefs), while the other half were asked to write beliefs denying sexual agency for women (low-agency beliefs). In this case, manipulation verification was ensured by participants providing any written response. Finally, participants completed scales measuring the dependent variables.

### 2.4. Materials

Socio-demographic Questionnaire. This section included questions about gender, age, nationality, sexual orientation, education, and relationship status, among others.

Identity Subscale of the Collective Self-Esteem Scale (ISE-CSES). Adapted from [33] ([33]), this scale measures the centrality of gender identity. Its four items are answered on a Likert scale ranging from 1 (strongly agree) to 7 (strongly disagree). In this study’s sample, the internal consistency reliability was α = 0.74.

Gender Self-Esteem Scale (GSES). Based on the work of [10] ([10]) and [11] ([11]), this scale evaluates the extent to which participants possess positive gender self-esteem. It comprises three items (e.g., “In general, I have a very high self-esteem as a woman”), answered on a Likert scale ranging from 1 (strongly disagree) to 7 (strongly agree). In this study’s sample, the internal consistency reliability was α = 0.60.

Neoliberal Beliefs and Values Questionnaire. Based on the Neoliberal Beliefs Inventory (NBI) ([6]), this instrument used 14 items rated on a Likert scale from 1 (strongly disagree) to 6 (strongly agree). These items correspond to two factors from the original scale: Competition (NBI1) (e.g., “Competitiveness is a good way to discover and motivate the most talented people”) and Personal Wherewithal (NBI2) (e.g., “A person’s success in life is more determined by their personal effort than by society”). Higher scores indicate a stronger endorsement of neoliberal beliefs. In this study’s sample, internal consistency reliability was α = 0.59 for NBI1 and α = 0.87 for NBI2.

Spanish Hetero-referred Version of the Sexual Double Standard Scale (SDSS-H). Adapted from [40] ([40]) and [22] ([22]), this scale measures perceived normativity, specifically the extent to which participants believe society accepts certain gender norms regarding sexual behavior. The scale comprises 18 items rated on a 4-point Likert scale from 0 (strongly disagree) to 3 (strongly agree), divided into three factors: Social Acceptance of Male Sexual Shyness (SAMSS), Social Acceptance of Female Sexual Freedom (SAFSF), and Social Acceptance of Sexual Double Standard (SASDS). In this study’s sample, internal consistency reliability was α = 0.65, α = 0.50, and α = 0.88 for SAMSS, SAFSF, and SASDS, respectively.

Self-Perception Measure. This measure consists of a list of 32 characteristics that assess the extent to which participants believe these traits describe them, rated on a 6-point Likert scale from 1 (not at all) to 6 (completely). These characteristics are grouped into eight theoretical dimensions. Six dimensions are based on the Stereotype Content Model ([15]; [14]) and procedures used by [20] ([20]): Competence (e.g., intelligence) and Sociability (e.g., generosity); Feeling (e.g., optimism) and Emotion (e.g., suffering) ([30]); Nature (e.g., intuition) and Culture (e.g., creativity) ([43]). Additionally, two extra dimensions were included: Individualism (e.g., autonomy) and Collectivism (e.g., solidarity). Each dimension included four characteristics, half with a positive connotation (e.g., Feeling: optimism) and half with a negative connotation (e.g., Feeling: remorse). Based on participants’ responses, eight Self-Assessment Indices (SAIs) were calculated to measure the tendency to identify with positive or negative traits within each dimension (COM: Competence, SOC: Sociability, CUL: Culture, NAT: Nature, FEEL: Feeling, EMO: Emotion, IND: Individualism, and COLL: Collectivism). To calculate each index (e.g., Self-Assessment Index for Competence: SAI-COM), the total sum of responses to negative traits (i.e., negative competence) was subtracted from the total sum of responses to positive traits (i.e., positive competence). Higher positive SAI scores indicate a stronger tendency to identify with positive traits over negative ones within a given dimension. Independent of dimension, additional scores were calculated for the overall tendency to identify with positive traits (POSIT) and negative traits (NEGAT).

Spanish Adaptation of the Sexual Consent Scale-Revised (SCS-R). Adapted from [24] ([24]) by [37] ([37]), the Spanish version consists of 26 items rated on a 7-point Likert scale (1 = completely disagree to 7 = completely agree). The scale includes four factors: Perceived Lack of Behavioral Control, Positive Attitude Toward Establishing Consent, Indirect Behavioral Approach, and Sexual Consent Norms. For this study, only two factors were analyzed: Perceived Lack of Behavioral Control (F1) and Attitude in Favor of Establishing Sexual Consent (F2). Internal consistency reliability for this study’s sample was α = 0.92 for F1 and α = 0.89 for F2.

### 2.5. Data Analysis

Three Multivariate Analyses of Covariance (MANCOVA) were conducted. In each analysis, two fixed factors were included: (1) Threat to Personal Reputation (with two levels: Sexual Activity and Sexual Abstinence) and (2) Affirmation in Agency Beliefs (with two levels: high-agency beliefs and low-agency beliefs). Covariates included scores on the ISE-CSES, GSES, NBI1, and NBI2 scales, as well as perceptions of societal support for gendered sexual norms —i.e., Social Acceptance of Male Sexual Shyness (SAMSS), Social Acceptance of Female Sexual Freedom (SAFSF), and Social Acceptance of the Sexual Double Standard (SASDS).

Regarding the dependent variables, the first MANCOVA included POSIT and NEG. In the second MANCOVA, the dependent variables were the eight Self-Assessment Indices (SAIs) corresponding to each dimension (COM: Competence, SOC: Sociability, CUL: Culture, NAT: Nature, FEEL: Feeling, EMO: Emotion, IND: Individualism, and COLL: Collectivism). In the third MANCOVA, the dependent variables were the two factors of the sexual consent scale: Perceived Lack of Behavioral Control and Attitude in Favor of Establishing Sexual Consent. Partial eta-squared (η_p_^2^) was used to quantify effect sizes.

The normality of distributions was verified using the Shapiro–Wilk test to ensure the suitability of parametric analyses. All analyses were conducted using SPSS version 28.

## 3. Results

The results of the marginal means for each condition and each variable can be found in Table 1.

### 3.1. Effect of Threat to Personal Reputation and Agency Beliefs on Identification with Positive and Negative Characteristics

The MANCOVA results did not reveal a significant effect of the factor Threat to Personal Reputation on POSIT and NEGAT (*F* = 2.07; *p* = 0.13; η_p_^2^ = 0.038). However, there was a significant effect of the factor Affirmation in Agency Beliefs (*F* = 3.38; *p* = 0.038; η_p_^2^ = 0.060) on positive and negative characteristics. No interaction effect was observed between the two factors (*F* = 1.40; *p* = 0.25; η_p_^2^ = 0.026).

When analyzing the dependent variables separately, for POSIT, there was a significant between-subjects effect of Threat to Personal Reputation (*F* = 4.03; *p* = 0.047; η_p_^2^ = 0.037), with higher scores in the Threat for Sexual Activity condition compared to Threat for Sexual Abstinence. Additionally, there was a significant effect of Affirmation in Agency Beliefs on POSIT (*F* = 6.54; *p* = 0.012; η_p_^2^ = 0.058), with higher scores in the low-agency beliefs condition compared to high-agency beliefs. The latter factor showed a larger effect size. No significant effect of either factor was observed on NEGAT.

Although there was no interaction effect between the two study factors, pairwise comparisons showed a significant effect of Threat to Personal Reputation on POSIT only within the low-agency beliefs condition (*F* = 6.31; *p* = 0.014; η_p_^2^ = 0.056), with higher POSIT scores in the Sexual Activity condition compared to Sexual Abstinence. Similarly, a significant effect of Affirmation in Agency Beliefs was observed only within the Threat to Personal Reputation for Sexual Activity condition (*F* = 9.13; *p* = 0.003; η_p_^2^ = 0.079), with a larger effect size than in the previous case: POSIT scores were higher in the low-agency beliefs condition compared to high-agency beliefs. The POSIT results are represented in Figure 1.

No significant effect was found for NEGAT.

### 3.2. Effect of Threat to Personal Reputation and Agency Beliefs on Self-Assessment Indices (SAIs)

The multivariate results of the MANCOVA did not reveal a significant joint effect on the eight Self-Assessment Indices for either the factor Threat to Personal Reputation (*F* = 0.99; *p* = 0.45; η_p_^2^ = 0.074) or the factor Affirmation in Agency Beliefs (*F* = 1.88; *p* = 0.070; η_p_^2^ = 0.132), although the latter approached significance. No interaction effect was found between the two factors (*F* = 1.01; *p* = 0.44; η_p_^2^ = 0.075).

The analysis of effects on each dependent variable revealed no significant between-subjects effects for the factor Threat to Personal Reputation on any dependent variable. However, the results approached significance for SAI-COLL (*F* = 3.50; *p* = 0.064; η_p_^2^ = 0.032) and SAI-COM (*F* = 3.67; *p* = 0.058; η_p_^2^ = 0.033), with higher scores in both cases for the Threat for Sexual Activity condition (CIs: [3.75–5.02] and [3.12–4.04] for SAI-COLL and SAI-COM, respectively) compared to the Threat for Sexual Abstinence condition (CIs: [2.87–4.16] and [2.46–3.40] for SAI-COLL and SAI-COM, respectively). Conversely, significant between-subjects effects of Affirmation in Agency Beliefs were observed for SAI-SOC (*F* = 4.19; *p* = 0.043; η_p_^2^ = 0.038), SAI-IND (*F* = 7.06; *p* = 0.009; η_p_^2^ = 0.062), and SAI-CUL (*F* = 4.04; *p* = 0.047; η_p_^2^ = 0.037). In all cases, higher scores were obtained in the low-agency beliefs condition compared to the high-agency beliefs condition, with the largest effect size observed for SAI-IND. Marginal means for these effects are presented in Table 1.

An interaction effect between Affirmation in Agency Beliefs and Threat to Personal Reputation was found for SAI-NAT (*F* = 6.34; *p* = 0.013; η_p_^2^ = 0.056). This interaction is explained by the presence of an effect of Affirmation in Agency Beliefs (i.e., higher scores in the low-agency Beliefs condition compared to the high-agency Beliefs condition) on SAI-NAT only under the Sexual Activity condition of Threat to Personal Reputation (*F* = 5.50; *p* = 0.021; η_p_^2^ = 0.049). This effect was not significant in the Sexual Abstinence condition of Threat to Personal Reputation (see Figure 2).

Similarly, although no significant interaction effect was found (*F* = 1.43; *p* = 0.23; η_p_^2^ = 0.013), pairwise comparison analyses revealed an effect of Affirmation in Agency Beliefs on SAI-CUL only under the Sexual Activity condition of Threat to Personal Reputation (*F* = 5.23; *p* = 0.024; η_p_^2^ = 0.047). Higher scores were also observed for the low-agency beliefs condition compared to the high-agency beliefs condition. No significant effect was found under the Threat for Sexual Abstinence condition.

### 3.3. Effect of Threat to Personal Reputation and Agency Beliefs on Attitude Toward Sexual Consent

The results of the MANCOVA revealed a significant interaction effect of the factors Threat to Personal Reputation × Affirmation in Agency Beliefs on the Attitude in Favor of Establishing Sexual Consent dimension (*F* = 3.23; *p* = 0.044; η_p_^2^ = 0.064). No main effects were found for the factor Threat to Personal Reputation (*F* = 1.60; *p* = 0.21; η_p_^2^ = 0.033) or the factor Affirmation in Agency Beliefs (*F* = 0.82; *p* = 0.44; η_p_^2^ = 0.017).

Specifically, a between-subjects interaction effect of the two factors was observed on the Attitude in Favor of Establishing Sexual Consent dimension of the SCS (*F* = 6.15; *p* = 0.015; η_p_^2^ = 0.060) (see Figure 3). More specifically, there was a significant effect of Affirmation in Agency Beliefs, with higher scores in the low-agency beliefs condition compared to the high-agency Beliefs condition, but only under the Threat to Personal Reputation for Sexual Activity condition (*F* = 7.27; *p* = 0.008; η_p_^2^ = 0.070).

Additionally, an effect of Threat to Personal Reputation was found, with higher scores for the Sexual Activity condition compared to the Abstinence condition, but only for the low-agency beliefs condition (*F* = 4.04; *p* = 0.047; η_p_^2^ = 0.040). No significant differences were observed for the Perceived Lack of Behavioral Control dimension of the SCS.

## 4. Discussion

This study assumes that in contemporary Western societies, women’s sexual behaviors and attitudes are regulated by a complex normative context resulting from two normative dimensions. On the one hand, the norm derived from the sexual double standard (SDS), which governs behaviors related to sexual activity and abstinence, promotes a negative evaluation of women who are assertive and sexually active ([2]; [9]; [28]; [51]). On the other hand, the norm of sexual agency, more recent in Western democratic societies, encourages young women to assertively express their own sexuality, opposing submission to traditional gender roles ([4]; [19]).

From this framework, and using a quasi-experimental design, we manipulated two factors [Threat to Personal Reputation: for sexual activity vs. for sexual abstinence, and Affirmation in Agency Beliefs: high-agency beliefs vs. low-agency beliefs] to study, first, the effect on participants’ identification with positive and negative traits that constitute the dimensions of stereotypes content ([14]; [43]; [57]). Secondly, we examined the effect of this complex normative context on young women’s attitude toward sexual consent ([61]).

### 4.1. How Did Threat to Personal Reputation Affect Participants?

We hypothesized that fear of confirming negative stereotypes might lead to the internalization of stereotypical expectations and roles, and, thus, inducing a threat to personal reputation could negatively impact women’s self-image ([23]; [52]). Specifically, we questioned how young women would be affected when exposed to a threat to their personal reputation via negative evaluation of their supposed sexual profile (i.e., characterized by sexual activity or sexual abstinence). We expected this situation to affect both their tendency to identify with positive and negative characteristics related to the dimensions of stereotype content ([14]) (H1) and their attitude toward sexual consent ([61]) (H2).

The first hypothesis is partially confirmed. We found that Threat to Personal Reputation has a significant effect on the tendency of participants to identify with positive traits across all dimensions, with this tendency being more pronounced when the Threat to Personal Reputation is for sexual activity (vs. sexual abstinence). At the limit of significance, girl participants whose reputations were threatened due to sexual activity, compared to those whose reputation was devalued due to sexual abstinence, tended to identify more with positive traits than with negative traits corresponding to the competence and collectivism dimensions.

The competence dimension, also referred to as “masculinity” or “instrumentality” ([14]; [62]), involves qualities relevant for achieving individual goals and objectives ([1]). Previous studies have shown that self-perception in terms of positive valence traits and competence is related to self-enhancement motives ([62]) and contributes to projecting an image of being capable of overcoming challenges ([42]). Additionally, within the framework of the Stereotype Content Model (SCM) ([8]; [15]; [14]), groups are perceived as more competent when they have high status and power. Thus, these findings suggest that the devaluation of young women’s personal reputation due to sexual activity prompts a tendency to emphasize their competence-related agency ([1]), potentially as a strategy to reaffirm their social status ([8]; [14]).

Self-descriptions in terms of collectivism–individualism are culturally influenced, with people in Western cultures having an independent self-view, emphasizing separation from others ([35]). The sample in this study consisted of young Spanish women, belonging to a Western society. Nonetheless, albeit at the border of statistical significance, girl participants whose reputations were devalued due to sexual activity (vs. sexual abstinence) tended to describe themselves in collectivist terms, highlighting their connection to others and an interdependent self-concept.

We found no support for our H2, as no significant main effect of Threat to Personal Reputation was observed on attitudes toward sexual consent.

### 4.2. How Did Affirmation in Agency Beliefs Affect Participants?

The agency norm reinforces the belief in free will and personal responsibility ([4]). We hypothesized that affirming high-agency beliefs (vs. low-agency beliefs) might enhance expectations of control and responsibility over the consequences of one’s behavior ([59]).

Our results partially support H3. We found that young women participants who affirmed that women’s range of sexual freedom is low (i.e., low-agency beliefs) were more likely than those who affirmed high-agency beliefs, to identify with positive traits across all dimensions of the stereotype, especially with positive rather than negative traits in the dimensions of *sociability, individualism,* and *culture*.

The sociability dimension, also referred to as “communion”, “warmth”, “femininity”, or “morality” ([14]; [62]), involves qualities relevant to social group belonging ([1]). Sociable individuals create an impression of warmth and morality in others ([14]; [42]).

The *nature–culture* dimension underpins a social classification where certain minority groups are excluded from the social map (ontologization) ([43]). While culture defines human identity, nature is associated with animal identity. Evidence suggests that women may be undervalued by being associated with traits that are less relevant in contexts of power, such as nature-related characteristics (e.g., maternal, intuitive) rather than culturally valued ones ([20]; [43]).

Women participants reaffirmed in low-agency beliefs also expressed a greater tendency, than those reaffirmed in high-agency beliefs, to identify with positive individualistic traits. Given the common national and cultural background of the sample, it seemed logical to expect no differential effect of manipulated factors on self-descriptions in terms of collectivism–individualism ([35]). However, the study design does not allow for conclusions beyond the obvious: when participants are reaffirmed in low-agency beliefs and, thus, made aware of the pressure of gender norms and stereotypes, they emphasize separation from others and traits of uniqueness.

### 4.3. How Do Threat to Personal Reputation and Affirmation in Agency Beliefs Interact?

To what extent can agency beliefs (i.e., low-agency vs. high-agency) moderate the threat to a woman’s sexual reputation? If stereotype-threatening situations ([53]) induce dissonance, can the effects of the threat to personal reputation be reduced by inducing beliefs that alter the balance between dissonant elements or ideas? ([47]) (H5). The tendency of the participating women to describe themselves with positive traits in any of the dimensions was greater when the threat to their personal reputation was due to sexual activity (vs. sexual abstinence) only in the condition in which they were induced to write low-agency beliefs, but this difference was not found when participants reaffirmed their beliefs in favor of high-agency. Furthermore, women participants who affirmed low-agency beliefs, compared to the group that affirmed high agency, showed a greater tendency to identify with positive traits of the dimensions corresponding to nature and culture, but this effect was only found in threat to reputation due to sexual activity condition but not in the condition “threat to reputation due to sexual abstinence (H5).

Finally, we found an interaction effect between the manipulated factors on attitude in favor of establishing sexual consent (H5). Specifically, attitudes toward sexual consent were more favorable when personal reputation was threatened due to sexual activity compared to abstinence but only when participants were reaffirmed in low-agency beliefs. Similarly, attitudes toward sexual consent were more favorable when participants affirmed low-agency beliefs (vs. high-agency beliefs) only if their personal reputation was threatened or devalued due to sexual activity.

Taken together, our results can be interpreted within the framework of theories on identification processes. First, from the perspective of Social Identity Theory (SIT) ([54]) and Self-Categorization Theory (SCT) ([55]), a threat to identity leads to a shift from personal identity to social identity, prompting individuals to self-define as members of the group. From this perspective, it can be inferred that the young women who participated in this study felt more threatened when their reputation was devalued for having an active sexual profile than for having an abstinent sexual profile. Moreover, according to SIT and SCT, we can assume that reaffirming high- or low-agency beliefs may facilitate a shift from individual identity (i.e., in the context of high-agency beliefs) to social identity (i.e., in the context of low-agency beliefs). This conclusion is based on the fact that women reaffirmed in low-agency beliefs (vs. those reaffirmed in high-agency beliefs) showed a greater tendency to describe themselves with positive traits and a more favorable attitude toward sexual consent.

A different issue is the variability observed in the dimensions of the stereotype con-tent that women chose to describe themselves. Regarding these results, we believe that future research should examine optimal identification strategies (i.e., assimilation or differentiation from the group) ([29]) that women choose in situations where their sexual reputation is threatened. Since there is evidence that the content of stereotypes is specific to female subtypes ([20]; [27]), it seems important to investigate to what extent the content of identification with these subtypes varies in order to satisfy the needs for assimilation or differentiation with the ingroup in situations of threat to sexual identity. Finally, future research should examine whether, when women are subjected to slut-shaming, reaffirming low-agency beliefs (vs. reaffirming high-agency beliefs) leads to lower perceptions of control, responsibility, guilt, or dissonance.

### 4.4. Limitations and Perspectives

There is still work to be conducted to understand how women’s self-concept and sexual attitudes may be influenced by the conflicting expectations stemming from the sexual double standard norm and the agency norm, which shape the normative gender context in many democratic Western societies.

This study used a threat to personal reputation, based on the sexual profile provided to the female participants, to analyze women’s self-concept. Therefore, strictly speaking, our findings cannot be interpreted as the effects of stereotype threat on women’s gender self-concept. Future research should explore whether the sexual double standard norm and the agency norm influence women’s awareness of gender stigma.

One relevant aspect to consider is how dominant coercive discourses regarding sexuality and consent might reinforce structural gender asymmetries, further complicating women’s ability to construct an autonomous and positive sexual self-concept. Recent contributions highlight how these discourses not only influence personal beliefs but also shape social interventions and public awareness campaigns ([44]). Future research should examine to what extent exposure to these coercive narratives moderates the effects of sexual agency norms and stereotype threats on self-perception and sexual attitudes. Understanding how these discourses operate could contribute to designing interventions that not only promote sexual consent awareness but also challenge the persistence of gender constraints in sexual reputation.

Some limitations of this research concern generalizability. The participants were mostly university students in their twenties, who generally have a different mindset compared to older adults, as the latter were socialized in a society with different gender values and norms. As a result, these findings may be limited to the population of university students. Further studies with more heterogeneous samples are needed, not only in terms of age but also by including non-heteronormative samples.

## 5. Conclusions

This study investigates the influence of normative contexts related to sexual gender norms on young women’s self-perceptions and on their attitude toward sexual consent. Our results indicate that normative contexts related to the sexual double standard and the sexual agency norm significantly shape both the valence and the nature of the attributes and characteristics that young women use to define themselves. Furthermore, the interaction between these two normative dimensions influences young women’s attitudes toward sexual consent. Several conclusions can be drawn from our results.

First, the tendency to reinforce a positive self-image is stronger when threats to personal reputation arise from sexual activity or assertiveness than when they originate from a profile of sexual abstinence or shyness. That is, expectations derived from the traditional version of the sexual double standard that regulates the domain of sexual freedom, rather than the version that regulates the domain of sexual abstinence, seem to activate the need for positive self-presentation in young women.

Second, this inclination toward positive self-description is more pronounced among young women who reaffirm the belief that women’s range of freedom is constrained by gender stereotypes, compared to those who reaffirm high-agency beliefs. Therefore, it seems that the sexual agency norm can hinder awareness or perception of the sociostructural and psychosocial barriers that continue to regulate gender roles.

Third, we found that norms derived from the sexual double standard and agency norms interact with each other. Specifically, when reputation is threatened by sexual activity, the tendency to describe oneself favorably is more pronounced than when reputation is devalued for being abstinent. However, this tendency of self-enhancement only occurs when young women reinforce the belief that their sexual activity does not depend on their dispositional characteristics but is, however, regulated by gender stereotypes and norms. This effect was observed specifically in the tendency to self-ascribe more positive than negative traits of the nature and culture domains. We found a similar interaction effect on favorable attitudes toward sexual consent. Women participating in this study showed a more favorable attitude toward sexual consent when their reputation was devalued for being sexually assertive rather than abstinent, but this effect depended on the reinforcement of low-agency beliefs. Attitudes toward sexual consent were also found to be more favorable among participants who reported a belief in low agency compared to those who reported a belief in high agency, but only when their reputation was undermined due to sexual activity.

Fourth, the results regarding the interaction between SDS-derived norms and the sexual agency norm seem to indicate that if a young woman’s sexual reputation is devalued for being assertive, she needs to be aware of the ongoing psychosocial asymmetry in relation to gender in order to deploy strategies that safeguard their self-concept, as well as to maintain a consistent and favorable attitude towards sexual consent.

In line with previous research that proposes counteracting the effects of dominant coercive discourses on women’s sexual self-perception and their attitudes toward consent ([44]), we suggest intervening in the following aspects. Providing information on how the sexual double standard operates can help young women become more aware of the sexist barriers that still exist in democratic societies. Additionally, informing young women about how the sexual agency norm functions can encourage a more conscious attitude toward the consequences of adhering to or complying with this norm without reflective consideration. We propose that such an intervention will promote greater self-determination among young women and may be effective in mitigating the harmful consequences of situations in which their sexual reputation is threatened, as well as a consistent, non-ambivalent attitude towards situations that require sexual consent.

## Figures and Tables

**Figure 1 behavsci-15-00250-f001:**
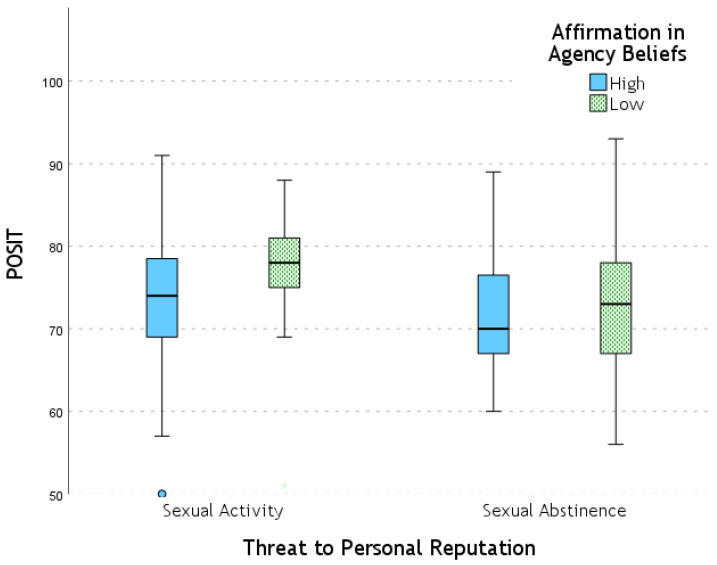
POSIT (positive traits) values across experimental conditions.

**Figure 2 behavsci-15-00250-f002:**
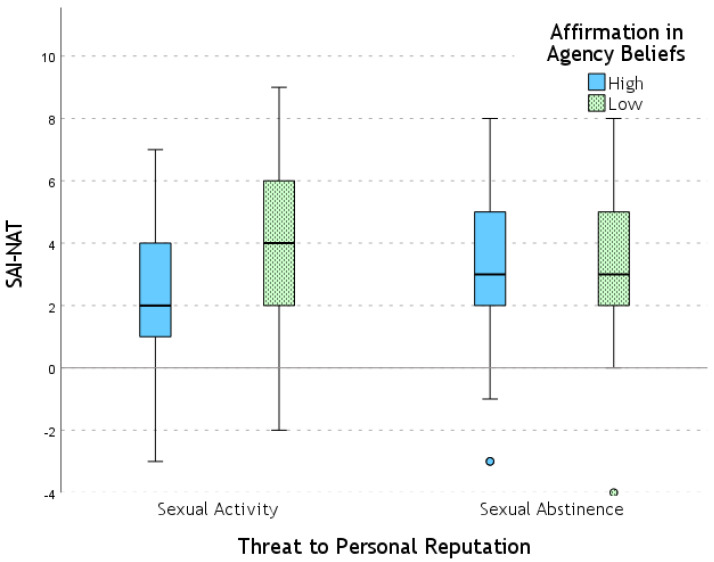
Self-Assessment Index values for the Nature dimension (SAI-NAT) across experimental conditions.

**Figure 3 behavsci-15-00250-f003:**
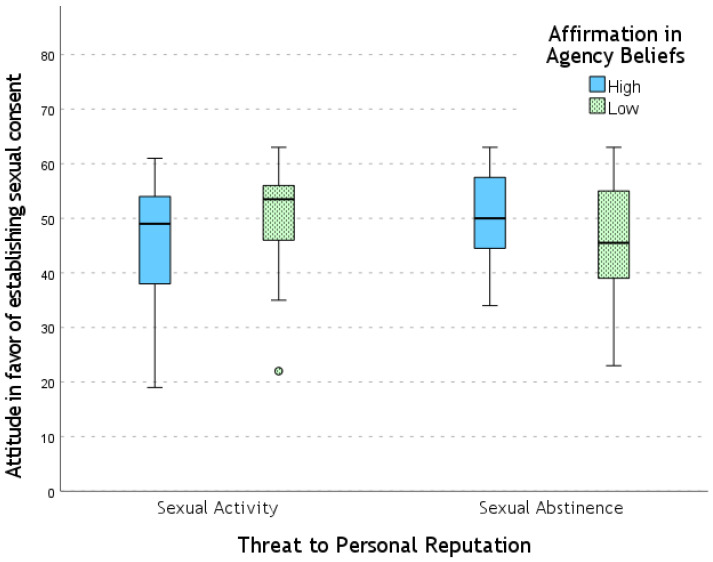
Values for the “Attitude Toward Establishing Sexual Consent” dimension of the SCS across experimental conditions.

**Table 1 behavsci-15-00250-t001:** Means and Standard Deviations (*M* ± *SD*) of Dependent Variables Across Experimental Conditions.

Dependent Variables	Threat to Personal Reputation	Affirmation in Agency Beliefs	
		Low	High
POSIT	Sexual Activity	76.5 ± 6.90	71.67 ± 8.24
	Sexual Abstinence	72.14 ± 8.28	71.03 ± 9.11
NEGAT	Sexual Activity	47.03 ± 7.87	47.88 ± 9.17
	Sexual Abstinence	47.17 ± 8.81	48.85 ± 8.79
SAI–COM	Sexual Activity	3.93 ± 0.35	3.22 ± 0.303
	Sexual Abstinence	3.07 ± 0.33	2.78 ± 0.34
SAI–SOC	Sexual Activity	3.84 ± 0.51	2.94 ± 0.44
	Sexual Abstinence	4.18 ± 0.48	3.13 ± 0.49
SAI-CUL	Sexual Activity	4.75 ± 0.46	3.35 ± 0.39
	Sexual Abstinence	3.51 ± 0.44	3.17 ± 0.44
SAI-NAT	Sexual Activity	3.81 ± 0.49	2.27 ± 0.42
	Sexual Abstinence	2.52 ± 0.46	3.38 ± 0.47
SAI-FEEL	Sexual Activity	2.18 ± 0.64	1.52 ± 0.55
	Sexual Abstinence	1.54 ± 0.60	1.46 ± 0.62
SAI-EMO	Sexual Activity	2.77 ± 0.51	2.98 ± 0.43
	Sexual Abstinence	2.18 ± 0.47	2.22 ± 0.49
SAI-IND	Sexual Activity	4.43 ± 0.48	3.28 ± 0.41
	Sexual Abstinence	3.77 ± 0.45	2.52 ± 0.46
SAI-COLL	Sexual Activity	4.83 ± 0.49	3.93 ± 0.42
	Sexual Abstinence	3.66 ± 0.46	3.36 ± 0.47
Perceived lack of behavioral control	Sexual Activity	21.28 ± 12.05	25.34 ± 12.39
	Sexual Abstinence	19.70 ± 10.89	18.95 ± 9.22
Attitude in favor of establishing sexual consent	Sexual Activity	52.08 ± 7.33	45.06 ± 11.83
	Sexual Abstinence	47.37 ± 10.21	51.04 ± 7.93

Note. SAIs: Self-Assessment Indices; COM: Competence, SOC: Sociability, CUL: Culture, NAT: Nature, FEEL: Feeling, EMO: Emotion, IND: Individualism, COLL: Collectivism, POSIT: positive traits, NEGAT: negative traits.

## Data Availability

The data will be available from the authors upon reasonable request and with the permission of the University of Granada.

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
