# Peer review of "Sexual and Agency Norms: Effect on Young Women’s Self-Perception and Attitude Toward Sexual Consent"

_behavsci, 2025, doi:10.3390/bs15030250_

Round 1
Reviewer 1 Report
Comments and Suggestions for Authors
The main aim of the research is to examine the influence of a normative environment of contemporary Western societies, on young women’s self-image and their attitude toward sexual consent. This topic is both original and highly relevant to the field. The paper effectively addresses a specific gap concerning sexual consent, particularly when it is devalued due to sexual assertiveness rather than abstinence. The paper successfully addresses a gap in the literature regarding the development of these variables within the Spanish social context, utilizing adapted measurement instruments such as the SDSS-H and SCS-R.
However, the introduction appears to be somewhat lacking in content.
The following topics should be explored in more depth:
- Include a definition of “sexual consent” and how the issue is addressed in the Spanish context (whether it is addressed, whether it is taboo, women are informed, etc.).
- What happens when a woman is attributed with a promiscuous sexual reputation? Discuss the effects. Additionally, it is recommended to include the following references:
Bay-Cheng, L. Y. (2019). Agency is everywhere, but agency is not enough: A conceptual analysis of young Women’s sexual agency. Journal of Sex Research, 56(4–5), 462–474. https://doi.org/10.1080/ 00224499.2019.1578330
Bean, T., Danh, L. T. M., Van, V. P., Thuy, N. T. T., Tam, D. N. H., & Thanh, P. T. T. (2024). Vietnamese Youth’s Perception on Slut-Shaming on Social Media. Trends in Psychology, 1-19.
Fasoli, F., Carnaghi, A., & Paladino, M. P. (2015). Social acceptability of sexist derogatory and sexist objectifying slurs across contexts. Language Sciences, 52, 98–107. https://doi.org/10.1016/j.langs ci.2015.03.003
Delgado Amaro, H., Alvarez, M. -J., & Ferreira, J. A. (2022). Sexual gender roles and stereotypes and the sexual double standard in sexual satisfaction among Portuguese college students: An exploratory study. Psychology & Sexuality, 1–16. https://doi.org/10.1080/19419899.2022.2039271
Kubota, S., & Nakazawa, E. (2024). Concept and implications of sexual consent for education: a systematic review of empirical studies. Sexual and Relationship Therapy, 39(3), 1021-1043.
Miano, P., & Urone, C. (2024). What the hell are you doing? A PRISMA systematic review of psychosocial precursors of slut-shaming in adolescents and young adults. Psychology & Sexuality, 15(1), 97-113. https://doi.org/10.1080/19419899.2023.2213736
Urone, C., Passiglia, G., Graceffa, G., & Miano, P. (2024). Pathways of self-determination: A constructivist grounded theory study of slut-shaming vulnerability in a group of young adults. Sexuality & Culture, 28(4), 1339-1368. https://doi.org/10.1007/s12119-023-10180-1
The conclusions are consistent with the evidence and arguments presented in the paper. They effectively address the main question posed and align with the study's objectives and findings. Moreover, it is recommended to include advice in the conclusions regarding the integration of sexual consent and violence prevention education into school curricula. This addition would enhance the practical implications of the study, addressing an essential aspect of applied intervention and aligning with broader societal efforts to promote healthy interpersonal relationships and reduce violence.
The references are appropriate; however, incorporating more recent sources would strengthen the work and provide greater relevance to the current context of the field.
The tables and figures are clear, well-organized, and effectively support the data and findings presented in the paper. No further improvements are needed in this area.
In general, I recommend expanding the theoretical background and adding more coherent references.
Author Response
Reviewer. Include a definition of “sexual consent” and how the issue is addressed in the Spanish context (whether it is addressed, whether it is taboo, women are informed, etc.)
Authors: Thank you for your suggestion for improvement. We have now included a definition of sexual consent, as well as its approach in the Spanish context (see page 3, paragraph 3).
Reviewer: What happens when a woman is attributed with a promiscuous sexual reputation? Discuss the effects. Additionally, it is recommended to include the following references:
Authors: Thank you for your suggestion. We have incorporated information in the 'Introduction' section about the consequences of a woman being attributed with a promiscuous sexual reputation, addressing its effects within the framework of the hypotheses we test in our study. Additionally, we have included some of the references suggested by the reviewer, as well as others we consider relevant to strengthen the analysis. Both the new information and the references have been highlighted in red in the text.
Reviewer. The conclusions are consistent with the evidence and arguments presented in the paper. They effectively address the main question posed and align with the study's objectives and findings. Moreover, it is recommended to include advice in the conclusions regarding the integration of sexual consent and violence prevention education into school curricula. This addition would enhance the practical implications of the study, addressing an essential aspect of applied intervention and aligning with broader societal efforts to promote healthy interpersonal relationships and reduce violence.
Authors. Thank you for your valuable suggestion. We have incorporated a discussion in the conclusions section addressing the integration of sexual consent and violence prevention education into school curricula. Specifically, we have included an analysis of how dominant coercive discourses regarding sexuality and consent can reinforce structural gender asymmetries, influencing both personal beliefs and public interventions. We highlight the need for future research to examine the moderating role of these discourses in shaping self-perception and sexual attitudes (see Limitations and Perspectives, paragraph 3). Additionally, in response to your recommendation, we propose specific intervention strategies aimed at counteracting the effects of coercive discourses on women's sexual self-perception and attitudes toward consent. We emphasize the importance of educating young women about the sexual double standard and the influence of sexual agency norms, promoting greater self-determination and a non-ambivalent attitude toward sexual consent (see Conclusion, paragraph 6). We believe these additions strengthen the practical implications of our study and align with broader societal efforts to promote healthy interpersonal relationships and reduce violence. We appreciate your insightful feedback, which has helped us refine the applied contributions of our work.
Reviewer. The references are appropriate; however, incorporating more recent sources would strengthen the work and provide greater relevance to the current context of the field.
Authors. Done. We have included new bibliographic sources and have indicated them in the list of references
Reviewer. In general, I recommend expanding the theoretical background and adding more coherent references.
Authors. We appreciate the reviewer's suggestion. In response to this recommendation, we have expanded the theoretical background of the manuscript to include more relevant and coherent references that strengthen the theoretical framework. We have added the most recent and pertinent research to better contextualize the work. Additionally, we have carefully reviewed the cited references to ensure their coherence and relevance, ensuring they provide solid support for the points discussed in the article. We hope that this expansion of the theoretical background and the inclusion of new references improves the clarity and depth of the manuscript.
Reviewer 2 Report
Comments and Suggestions for Authors
The manuscript offers a methodologically sound and provocative analysis of how agency beliefs and sexual double standards interact to influence young women's attitudes on sexual consent and their perceptions of themselves. The subject is extremely pertinent, especially in light of the current discussion surrounding gender roles and sexuality, and the results advance our knowledge of the psychological effects of these competing social expectations.
The use of validated scales (such as the Spanish version of the Sexual Consent Scale-Revised) guarantees accurate assessment, and the quasi-experimental 2x2 design is ideal for examining interactions between Threat to Personal Reputation and Agency Beliefs. Furthermore, the statistical analyses are made more robust by the use of MANCOVA and the disclosure of effect sizes. The findings on self-enhancement under reputational threat and the relationship between sexual reputation depreciation and agency beliefs are noteworthy because they shed light on how young women deal with these social expectations. One particularly striking finding is that, solely under low-agency belief conditions, views about sexual consent were more positive when sexual activity (rather than abstention) harmed one's reputation. This finding merits deeper investigation.
Although the study's positive aspects, a few little changes might improve its rigor and clarity.
From a methodological perspective, the study does not indicate whether the manipulations were equally effective across conditions, but it does remove subjects based on manipulation checks. It would be easier to verify that the experimental conditions were interpreted as intended if a quick ANOVA on manipulation check answers was included. Furthermore, as subjects pertaining to sexual conduct and gender norms are especially vulnerable to social desirability bias, the use of self-report measures requires a more careful examination. Resolving this restriction would improve how the findings are interpreted.
In the results section, some borderline p-values (e.g., p = 0.058, p = 0.064) suggest trends rather than strong effects. Reporting confidence intervals or incorporating Bayesian analysis could provide a more nuanced interpretation. Furthermore, given the significant interaction effects, conducting a simple slopes analysis would clarify the nature of these interactions beyond the MANCOVA results.
In the discussion, the unexpected finding that low-agency beliefs enhanced self-perception in the sexual activity condition could be explored in greater depth. One possible explanation is that self-enhancement under reputational threat serves as a motivated self-protection strategy. Discussing this within the framework of identity negotiation theories would provide a richer theoretical context for the findings.
Comments on the Quality of English LanguageSeveral minor typographical errors and awkward phrasings should be corrected for readability and professionalism. For example:
- Title (Page 1): Please replace "Attitud Toward Sexual Consent" with "Attitude Toward Sexual Consent"
- Participants Section (Page 4): Please remove "of those" in the sentence "Among them, 43.5% reported being in a romantic relationship, and 79.1% of those identified as heterosexual."
- Results Section (Page 7): Please replace "Attitude toward sexual consent were more favorable..." with "Attitudes toward sexual consent were more favorable..."
- Conclusion (Page 13): Please replace "The dominio of sexual freedom..." with "The domain of sexual freedom..."
Author Response
Reviewer. One particularly striking finding is that, solely under low-agency belief conditions, views about sexual consent were more positive when sexual activity (rather than abstention) harmed one's reputation. This finding merits deeper investigation.
Authors. Thank you for highlighting this important finding. In response, we have expanded our discussion to further explore the implications of this result. Specifically, we interpret it within the framework of identification processes, suggesting that under reputational threat, reaffirming low-agency beliefs may serve as a self-protection mechanism, leading to more positive views on sexual consent. Additionally, we propose that future research should examine the role of optimal identification strategies (assimilation vs. differentiation) in shaping these attitudes and explore whether reaffirming low-agency beliefs influences perceptions of control, responsibility, or dissonance in slut-shaming contexts (see Discussion).
Reviewer: Although the study's positive aspects, a few little changes might improve its rigor and clarity.
From a methodological perspective, the study does not indicate whether the manipulations were equally effective across conditions, but it does remove subjects based on manipulation checks. It would be easier to verify that the experimental conditions were interpreted as intended if a quick ANOVA on manipulation check answers was included. Furthermore, as subjects pertaining to sexual conduct and gender norms are especially vulnerable to social desirability bias, the use of self-report measures requires a more careful examination. Resolving this restriction would improve how the findings are interpreted.
Authors. We thank the reviewer for these recommendations. However, we have the impression that we did not refer properly to the verification question we asked to ensure that participants had read the instructions. We called it as “manipulation check” in section “Participants”. This fact may have led to a misunderstanding. We did not make strictly a manipulation check, since we expect an implicit influence of the stimuli. Indeed, we agree with the reviewer, participants are usually vulnerable to social desirability bias and we tried to mitigate this effect by avoiding direct questions on the interpretation of the text. To avoid confusión, we have rephrased the sentence to be more precise as follows: “An additional 18 participants were excluded for not responding to the verification question that was used to confirm they had read the text on Affirmation in Agency Be-liefs (see the “Procedure” section).”
Reviewer. In the results section, some borderline p-values (e.g., p = 0.058, p = 0.064) suggest trends rather than strong effects. Reporting confidence intervals or incorporating Bayesian analysis could provide a more nuanced interpretation. Furthermore, given the significant interaction effects, conducting a simple slopes analysis would clarify the nature of these interactions beyond the MANCOVA results.
Authors. We thank the reviewer for these suggestions. We have the same interpretation of the results, they suggest trends rather than evident effects. Maybe these results would be different with a higher number of participants. Following the recommendation of the reviewer we have incorporated confidence intervals (CI) in section 3.2. for the p value results approaching signficance: “However, the results approached significance for SAI-COLL (F = 3.50; p = .064; ηp2 = .032) and SAI-COM (F = 3.67; p = .058; ηp2 = .033), with higher scores in both cases for the Threat for Sexual Activity condition (CIs: [3.75 – 5.02] and [3.12 – 4.04], for SAI-COLL and SAI-COM, respectively) compared to the Threat for Sexual Abstinence condition (CIs: [2.87 – 4.16] and [2.46 – 3.40], for SAI-COLL and SAI-COM, respectively).” After an exhaustive observation of the results, given the comments from other reviewers and given that interactions have been obtained with categorical variables, we are not sure how slope analysis could contribute to clarify the interactions in our case. Therefore, we would prefer to keep the associated results as presented as the contrasts involved in the interaction are also detailed. We keep in mind this suggestion for future studies including moderation effects and we will also consider the Bayesian approach.
Reviewer. In the discussion, the unexpected finding that low-agency beliefs enhanced self-perception in the sexual activity condition could be explored in greater depth. One possible explanation is that self-enhancement under reputational threat serves as a motivated self-protection strategy. Discussing this within the framework of identity negotiation theories would provide a richer theoretical context for the findings.
Authors. Thank you for your insightful suggestion. In response, we have expanded the discussion by framing our findings within theories of identification processes, particularly Social Identity Theory (SIT) and Self-Categorization Theory (SCT). We propose that reaffirming low-agency beliefs may facilitate a shift from individual to social identity, serving as a self-protection mechanism under reputational threat (see Discussion, page 12, paragraph 2). Additionally, we highlight the need for future research to explore optimal identification strategies (assimilation vs. differentiation) in response to sexual reputation threats and to examine whether reaffirming low-agency beliefs influences perceptions of control, responsibility, guilt, or dissonance in slut-shaming contexts (see Discussion, page 12, paragraph 3). We appreciate your feedback, which has helped us provide a richer theoretical context for our findings.
Reviewer. Several minor typographical errors and awkward phrasings should be corrected for readability and professionalism. For example:
- Title (Page 1): Please replace "Attitud Toward Sexual Consent" with "Attitude Toward Sexual Consent"
- Participants Section (Page 4): Please remove "of those" in the sentence "Among them, 43.5% reported being in a romantic relationship, and 79.1% of those identified as heterosexual."
- Results Section (Page 7): Please replace "Attitude toward sexual consent were more favorable..." with "Attitudes toward sexual consent were more favorable..."
- Conclusion (Page 13): Please replace "The dominio of sexual freedom..." with "The domain of sexual freedom..."
Authors. Thank you for your suggestion for improvement. We have corrected all typographical errors that you have pointed out
Reviewer 3 Report
Comments and Suggestions for Authors
This quasi-experimental study investigates the influence of normative contexts related to sexual gender norms on young women's self-perceptions and their attitudes toward sexual consent. The results indicate that normative contexts related to the Sexual Double Standard (SDS) and the Sexual Agency Norm significantly shape both the valence and nature of the attributes and characteristics that young women use to define themselves. Furthermore, the interaction between these two normative dimensions influences young women's attitudes toward sexual consent.
The manuscript appears clear and relevant to the field of gender studies and behavioral science, particularly in understanding the impact of sexual norms on young women's self-perception and attitudes toward sexual consent. The references include a wide range of prior knowledge. The experimental design, a 2x2 quasi-experimental design, seems appropriate to test the hypotheses regarding the impact of normative contexts on self-perception and attitudes. The hypotheses are clearly stated, and the methods are detailed enough to understand the experimental setup. Figures and tables are appropriate for displaying the data. The data is consistent and appropriately interpreted throughout the manuscript.
The importance of these findings lies in their potential to inform interventions aimed at addressing conflicts related to sexual consent. However, it is challenging to derive clear guidelines from these results that could contribute to guiding, for example, campaigns or interventions for the promotion of sexual consent free from coercion. This aspect should be improved.
For the discussion of the results, there are contributions on the impact of the dominant coercive discourse and sexual consent that would help to better understand the reasons behind these findings. These contributions can be translated into clear guidelines for interventions to overcome the dominant coercive discourse. I provide some references in this regard, which you are not obliged to include:
Cañaveras, P., De Botton, L., Carbonell, S., Elboj, C., Aubert, A., & Lopez de Aguileta, G. (2024). Youth Voices Participating in the Improvement of Sexual Consent Awareness Campaigns. Sexes 5, 579-595. https://doi.org/10.3390/sexes5040038
Racionero-Plaza, S., Puigvert, L., Soler-Gallart, M & Flecha, R. (2022). Contributions of Socioneuroscience to Research on Coerced and Free Sexual-Affective Desire. Frontiers in Behavioral Neuroscience. 15(814796). https://doi.org/10.3389/fnbeh.2021.814796
Author Response
Reviewer. The importance of these findings lies in their potential to inform interventions aimed at addressing conflicts related to sexual consent. However, it is challenging to derive clear guidelines from these results that could contribute to guiding, for example, campaigns or interventions for the promotion of sexual consent free from coercion. This aspect should be improved.
Authors. Thank you for your valuable suggestion. In response, we have expanded the discussion on the practical implications of our findings by proposing specific intervention strategies to counteract the effects of coercive discourses on women's sexual self-perception and attitudes toward consent. Specifically, we highlight the importance of integrating education on the sexual double standard and sexual agency norms into awareness campaigns and intervention programs. These strategies aim to foster greater self-determination among young women and promote a clear, non-ambivalent stance toward sexual consent (see Conclusion, paragraph 6). We believe these additions enhance the applicability of our study and contribute to broader societal efforts to promote healthy interpersonal relationships and reduce violence. We appreciate your insightful feedback, which has helped us strengthen the real-world impact of our research.
Reviewer. For the discussion of the results, there are contributions on the impact of the dominant coercive discourse and sexual consent that would help to better understand the reasons behind these findings. These contributions can be translated into clear guidelines for interventions to overcome the dominant coercive discourse. I provide some references in this regard, which you are not obliged to include:
-Cañaveras, P., De Botton, L., Carbonell, S., Elboj, C., Aubert, A., & Lopez de Aguileta, G. (2024). Youth Voices Participating in the Improvement of Sexual Consent Awareness Campaigns. Sexes 5, 579-595. https://doi.org/10.3390/sexes5040038
-Racionero-Plaza, S., Puigvert, L., Soler-Gallart, M & Flecha, R. (2022). Contributions of Socioneuroscience to Research on Coerced and Free Sexual-Affective Desire. Frontiers in Behavioral Neuroscience. 15(814796). https://doi.org/10.3389/fnbeh.2021.814796
Authors. Thank you for your suggestion for improvement. The authors have reviewed the recommended literature and we have considered it relevant to consider your suggestion, so we have included two new brief paragraphs in the discussion (limitations section and conclusions section). Thank you very much.
- Cañaveras, P., De Botton, L., Carbonell, S., Elboj, C., Aubert, A., & Lopez de Aguileta, G. (2024). Youth Voices Participating in the Improvement of Sexual Consent Awareness Campaigns. Sexes5, 579-595. https://doi.org/10.3390/sexes5040038
- Racionero-Plaza, S., Puigvert, L., Soler-Gallart, M & Flecha, R. (2022). Contributions of Socioneuroscience to Research on Coerced and Free Sexual-Affective Desire. Frontiers in Behavioral Neuroscience.15(814796). https://doi.org/10.3389/fnbeh.2021.814796